# Seasonal and Spatial Variability of Dissolved Nutrients in the Yenisei River

Irina V. Tokareva [1,*] and Anatoly S. Prokushkin [1,2]

1   V.N. Sukachev Institute of Forest SB RAS, Akademgorodok 50/28, 660036 Krasnoyarsk, Russia
2   Tomsk State University, Lenina Pr 36/13, 634050 Tomsk, Russia
*   Correspondence: gavrilenko@ksc.krasn.ru

**Abstract:** The accelerated rates of warming in high latitudes lead to permafrost degradation, enhance nutrient cycling and intensify the transport of terrestrial materials to the Arctic rivers. The quantitative estimation of riverine nutrient flux on seasonal and spatial scales is important to clarify the ongoing changes in land–ocean connectivity in the Arctic domain. This study is focused on a multiyear (2015–2021) analysis of concentrations of dissolved nutrients in the Yenisei River. Applying stationary water sampling, we studied seasonal variations in concentrations of phosphate, nitrate, nitrite and ammonia ions in the Yenisei River in the upper (56.0° N), middle (60.9° N) and lower (67.4° N) sections of the river. The waters of the upper river section demonstrated significant and steady nutrient enrichment throughout the hydrological year, reflecting the influence of the Krasnoyarsk reservoir. The downstream reaches of the Yenisei River had more apparent seasonal patterns of nutrient concentrations. Particularly, winter-season nutrient levels in the middle and lower river sections were the highest during the hydrological year and close to the upper section. At snowmelt, and especially the summer–fall seasons, all inorganic nutrient concentrations dropped dramatically after the inflow of the Angara River. On the other hand, the peak nitrite content observed during the early spring flood was most pronounced in the lower section of the river basin, reflecting the specific characteristics of the nitrogen cycle in permafrost soils. The spring flood plays the major role in the annual nutrient fluxes, except for nitrates, for which the maximum occurred in the winter season. The summer–fall season, despite its duration and considerable water runoff, demonstrated the lowest fluxes of dissolved inorganic phosphorus and nitrogen in comparison to other periods of the hydrological year, suggesting strong biological uptake and chemostasis.

**Keywords:** river runoff; river chemistry; Siberia; phosphate; nitrate; nitrite; ammonium

## 1. Introduction

The Yenisei River is among the largest rivers in Siberia, crossing several climatic zones and carrying significant annual fluxes of water and dissolved material into the Arctic Ocean [1–4] (Amon et al., 2012). Predicted climate warming [5,6] may lead to changes in the hydrothermal regime of soils and degradation of permafrost [7,8], increasing the rates of mineralization of soil organic matter accumulated in terrestrial ecosystems [9,10] and the weathering of parent rocks [11,12] and, accordingly, radically changing the biogeochemical balance of high-latitude ecosystems [11–18], including intensifying the transfer of nutrients, such as N and P, into the Arctic Ocean [17,19,20]. An equally important factor is changes in the hydrological regime (increased water runoff, seasonal flux distribution, nutrient sources, etc.) in the river basins of the Arctic domain [21–25], which imply the transformation of the hydrochemical composition of river waters entering the Arctic Ocean [18,26].

The nutrient regime in freshwater ecosystems is one of the most important factors regulating biological diversity and the ecosystem productivity of lotic and lenthic systems [27]. The relative concentrations of nitrogen and phosphorus and their ratios can function as important indices reflecting the nutrient limitations of primary producers in

aquatic ecosystems and overall freshwater ecosystem productivity [28]. Nitrogen is the major limiting nutrient in boreal and Arctic terrestrial ecosystems, which, without disturbances, usually minimize its losses to aquatic systems [29]. On the other hand, the actual N stock in permafrost-affected soils can reach 130 Pg (petagram) of N [30]. The potential higher availability of inorganic nitrogen in high-latitude terrestrial ecosystems might be a source enhancing nitrous oxide emissions to the atmosphere [20] and/or leaching to aquatic ecosystems [31]. The greatest N turnover occurs during the summer period, but winter N mineralization may provide an important N source upon thawing in spring, when, at snowmelt, a nitrate pulse occurs in the soil solution of Arctic ecosystems [32,33] and river water [34,35]. Enhanced phosphorus biogeochemical cycling in terrestrial ecosystems under a warmer climate can also affect the productivity of freshwater and, finally, marine ecosystems [36]. Therefore, permafrost degradation is expected to raise the riverine exports of phosphorus, nitrogen and suspended solids in Arctic rivers [37]. The nutrient input from land with Arctic rivers is a key process that will affect the future evolution of the Arctic Ocean, which receives around 11% of global freshwater discharge and the catchments of which are the fastest-changing regions due to anthropogenic climate change [38].

In recent decades, significant efforts have been made to asses quantitative parameters of nutrient loads to the Arctic Ocean with the runoff of great Arctic rivers; particularly, the Yenisei River [25,37,39–46]. However, these studies were limited by single-point sampling stations (e.g., terminal gauge stations), low-frequency sampling (three to five sampling occasions per year) and relatively short periods of observation (the longest series being during the PARTNERS/Arctic-GRO program). In this regard, an analysis of the current state of water resources in this region and a forecast of their changes are currently very important.

The purpose of our work was to assess the spatiotemporal dynamics of nutrient concentrations in the Yenisei River channel and the overall persistence of ecosystem processes and nutrient downstream transport from headwaters to the Kara Sea. The key objectives of the research were: 1) the analysis of spatial changes in the concentrations of major nutrients (nitrates, nitrites, ammonium and phosphates) in the Yenisei River, 2) an examination of seasonal fluctuations in nutrient concentrations during the hydrological year and 3) an evaluation of the annual export of biogenic elements from the Yenisei River basin into the Arctic Ocean.

## 2. Materials and Methods

### 2.1. River Basin

The Yenisei River is one of the largest rivers in the world and ranks first among the rivers in Russia in terms of water runoff. Its average long-term runoff is 636 km$^3$/year, its catchment area is 2.54 million km$^2$, its length is 4803 km and its basin extends in a latitudinal direction from 51°43′ N to 69°36′ N and covers several climate provinces [4]. The Yenisei River catchment, according to the physical, geographical conditions and the water regime, is divided into three sections: an upper section (the headwaters—mouth of the Angara River), middle section (Angara River—mouth of the Nizhnyaya Tunguska River) and lower section (Nizhnyaya Tunguska River—mouth of the Yenisei river) [47].

### 2.2. River Water Sampling and Dissolved Nutrient Measurements

Water sampling from the river channel was carried out from 2015 to 2021 weekly during the open water period (May–September) and monthly during the winter (October–April) in the middle part of the river channel at depths of 20 to 30 cm. The dissolved biogenic element concentrations in the Yenisei River were analyzed at four stationary sites (Figure 1), representing:

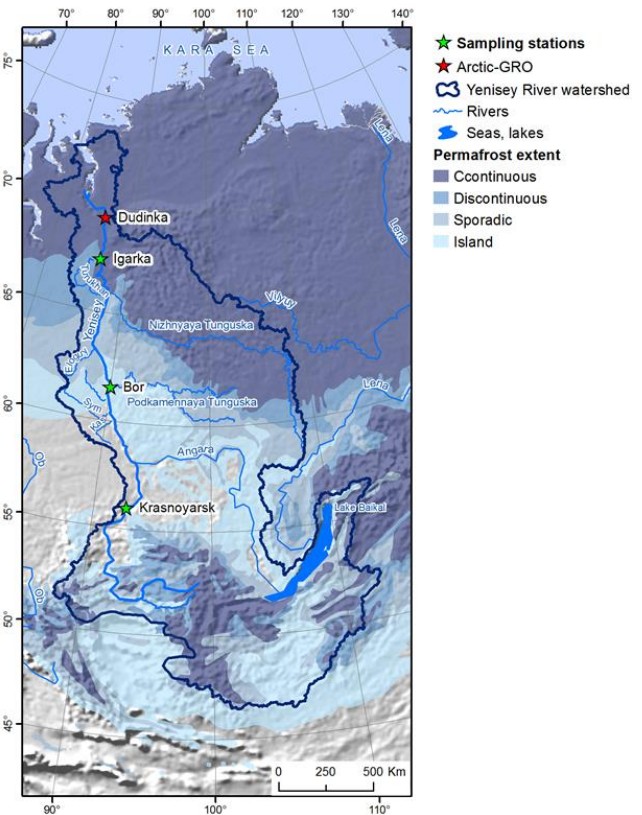

**Figure 1.** Drainage basin of the Yenisei River with sampling stations with a background map of the permafrost distribution adapted from Brown et al., 1998 [48].

1. The upper section: downstream of the dam of the Krasnoyarsk hydroelectric power station (Krasnoyarsk, 56.0° N, distance to outlet: 2468 km);

2. The middle section: near Zotino village (Zotino Tall Tower Observatory (ZOTTO), 60.9° N, distance to the outlet: 1568 km);

3. The lower section: near Igarka (Igarka Geocryology Laboratory of the Melnikov Permafrost Institute SB RAS, 67.5° N, distance to the outlet: 697 km);

4. The lower section: near Dudinka, data from the Arctic-GRO database (https://arcticgreatrivers.org/data/ accessed on 8 September 2022) (69.2° N, distance to the outlet: 304 km).

At the Igarka station, sampling was carried out with greater frequency (ca. every 5 days) from 2015 to 2021, which allowed a more detailed assessment of the temporal variability in nutrient concentrations during the hydrological year.

Immediately after collection, water samples were filtered using cellulose filters (0.22 μm, Millipore) and frozen at −18 °C until laboratory analysis. Nutrient determination methods are shown in Table 1.

**Table 1.** Nutrient analysis methods.

| Nutrient | Instrument | Method | Detection Limits |
|---|---|---|---|
| P-PO$_4$ | Flow injection analyzer Lachat Quikchem 8500 (Loveland, CO, USA) | 10-115-01-1-M | $\geq$0.1 µgP/L |
| N-NO$_3$ | Flow injection analyzer Lachat Quikchem 8500 (Loveland, CO, USA) | 10-107-04-1-O | $\geq$7 µgN/L |
| N-NO$_2$ | Flow injection analyzer Lachat Quikchem 8500 (Loveland, CO, USA) | 10-107-04-1-O | $\geq$1.5 µgN/L |
| N-NH$_4$ | Flow injection analyzer Lachat Quikchem 8500 (Loveland, CO, USA) | 10-107-06-5-H | $\geq$0.46 µgN/L |

*2.3. Discharge and Nutrient Flux*

The archive data on the mean daily water discharge of the Yenisei River at the gauging stations Bazaikha (Krasnoyarsk), Podkamennaya Tunguska (Bor) and Igarka were obtained from the Central Siberian Department of Hydrometeorology and Environmental Monitoring (Roshydromet). To calculate the average discharge for hydrological periods, we used available data for 2015 to 2018 and the average annual runoff values (Table 2) according to the data [49]. The data on daily discharges for Igarka station in 2019 to 2021 were obtained from the Arctic-GRO website (https://arcticgreatrivers.org/discharge/, accessed on 8 September 2022).

**Table 2.** Seasonal and annual discharge of the Yenisei River for 2015 to 2018 and long-term mean values at the Roshydromet gauging stations: Krasnoyarsk (Bazaikha, 56.0° N, 2468 km to the outlet), Bor (Podkamennaya Tunguska, 61.5° N, 1568 km) and Igarka (67.4° N, 697 km).

| Period | Bazaikha | Bor | Igarka |
|---|---|---|---|
| | Water discharge, m$^3$/s (% of annual flow) | | |
| Spring flood (2015–2018) | 2650 $\pm$ 200 (19) | 20,150 $\pm$ 1500 (36) | 46,530 $\pm$ 6620 (44) |
| Long-term mean | 3110 * | 25,780 ** | 53,220 *** |
| Summer–fall (2015–2018) | 2800 $\pm$ 60 (36) | 9690 $\pm$ 290 (34) | 16,400 $\pm$ 1070 (31) |
| Long-term mean | 3020 | 11,150 | 18,540 |
| Winter low flow (2015–2018) | 2150 $\pm$ 110 (45) | 5610 $\pm$ 230 (30) | 8940 $\pm$ 880 (25) |
| Long-term mean | 2490 | 6440 | 7020 |
| Annual (2015–2018) | 2660 | 9680 | 17,290 |
| Annual long-term mean | 2800 | 10,860 | 18,520 |

Notes: * averaged over the period 1967–2018 (after the filling of the reservoir), ** 1936–2018, *** 1936–2018.

The division of the hydrological year into periods (winter low-flow, spring flood and summer–fall seasons) was different for the selected sampling stations. For the southernmost Bazaikha station, the spring flood occurred in May–June, the summer–fall period was July–November and the winter period was December–April. For the Bor (Podkamennaya Tunguska) and Igarka stations, the average seasonal discharge in the spring flood was calculated as the average for May–June, the summer–fall period (July–October) and the winter period (November–April). The nutrient flux for the corresponding hydrological period was calculated on the basis of the averaged values of the ion concentrations in the samples collected at these stations in 2015–2021 and the water discharge obtained

as described above. The absolute error in the value of the nutrient fluxes for separate hydrological periods was determined according to the formula:

$$\sigma_f = (\sigma_c + \sigma_q) \times m_f, \tag{1}$$

where $\sigma_f$ is the absolute error of the element runoff value, $\sigma_c$ is the relative error of the mean seasonal concentration of the element, $\sigma_q$ is the relative error of the seasonal average water flow rate and $m_f$ is the product of the seasonal average concentration of the element and the corresponding water flow rate.

The nutrient yields were calculated as a nutrient flux (mean annual value) normalized to a specific drainage area for each sampling station (Bazaikha—287,679 km$^2$, Bor—1,764,853 km$^2$ and Igarka—2,437,106 km$^2$). Dissolved inorganic nitrogen (DIN) was calculated as the sum of the $NO_3$ and $NH_4$ concentrations. Nitrite was not included in the calculation due to the lack of data for the Dudinka station. The N:P ratio was estimated in terms of molar mass.

Statistical data processing was carried out using the Statistica 10 package.

## 3. Results

### 3.1. Hydrological Regime

According to the data from long-term observations, there is a 6.7 times increase in the water discharge of the Yenisei River from the border of Krasnoyarsk (Bazaikha) to the lowest section (Igarka), from 87 to 603 km$^3$. In the latitudinal direction, the seasonal runoff distribution changes significantly. Due to the regulated flow in the upper reaches, the share of the spring season in the annual flow is about 19%, but it reaches 44% in the lower section (Table 2), thus making it, despite its short duration, the main hydrological period. The period considered in our study (2015–2018) was characterized by a reduced annual runoff, which was expressed to a lesser extent in the upper reaches (96% of the long-term average value) but more apparent in downstream sections: in the Podkamennaya Tunguska section (89%) and Igarka (90%). The most significant decrease in the water discharge of the Yenisei River was observed during the spring flood period in the Podkamennaya Tunguska section (82% of the long-term average) and Igarka (87%). The reduced values of the flow rates at these stations were also observed during the summer–fall low-flow season (86–88%).

### 3.2. Nutrients in the Upper Section

Seasonal mean concentrations of phosphates in the waters of the Yenisei River near Krasnoyarsk varied from 2015 to 2021 within the range from 7 to 65 µgP/L. The maximum levels of phosphates (24.4 ± 16.9 µgP/L) occurred during the spring flood (Figure 2a). Lower values for the phosphate content were observed at the summer–fall (17.1 ± 11.9 µgP/L) and winter (17.5 ± 3.9 µgP/L) periods.

The nitrate nitrogen concentrations within the upper section of the Yenisei River had an insignificant intra-annual amplitude, ranging from about 160 µgN/L to 170 µgN/L in the summer–fall period and the spring and winter low-flow periods, respectively (Figure 2b). The other forms of inorganic nitrogen, however, demonstrated clear inter-seasonal changes: the minimum nitrite concentrations were observed in the spring flood (Figure 2c) and ammonium nitrogen reached the highest concentrations in winter (Figure 2d). The total amount of dissolved inorganic nitrogen (DIN) in river water at this station varied in the range from 205 to 226 µgN/L, and the N:P molar ratio was in the range from 20 to 27, depending on the hydrological season (Table 3).

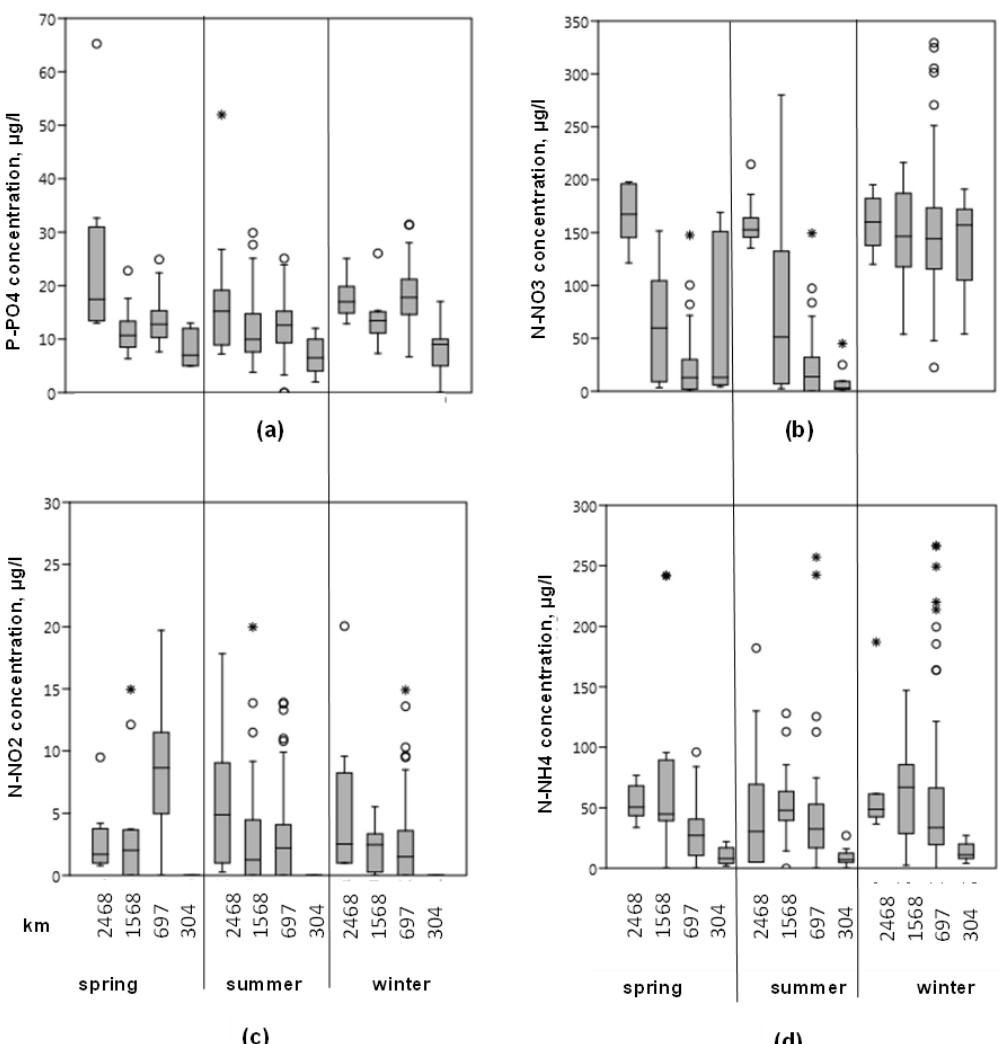

**Figure 2.** The nutrient content in the water of the Yenisei River at different latitudes and hydrological seasons: phosphate concentrations (**a**), nitrate concentrations (**b**), nitrite concentrations (**c**) and ammonium concentration (**d**). Numbers correspond to the distance to the outlet in km: 2468 km—Krasnoyarsk (56.0° N), 1568 km—Zotino (60.9° N), 697 km—Igarka (67.4° N), 304 km—Dudinka (69.2° N).

**Table 3.** Total dissolved inorganic nitrogen (DIN = $NO_3$ + $NH_4$) and the N:P molar ratio in the water of the Yenisei River in different hydrological periods in the longitudinal direction.

| Hydrological Period | Bazaikha | | Bor | | Igarka | | Dudinka * | |
|---|---|---|---|---|---|---|---|---|
| | DIN | N:P | DIN | N:P | DIN | N:P | DIN | N:P |
| Spring flood | 221 ± 37 | 20 | 149 ± 78 | 29 | 59 ± 39 | 10 | 81 ± 85 | 22 |
| Summer–fall | 210 ± 58 | 25 | 134 ± 81 | 23 | 72 ± 56 | 13 | 18 ± 15 | 6 |
| Winter low-flow | 231 ± 41 | 27 | 210 ± 62 | 33 | 199 ± 87 | 25 | 153 ± 44 | 39 |

Notes: * data from the Arctic-GRO database.

### 3.3. Nutrients in the Middle Section

The phosphate concentrations decreased in the middle section by 2.0 (spring) to 1.3 times (summer–fall and winter periods) compared to the upper section. A similar dynamic was observed for nitrate, except for the winter low-water period, when concentrations remained similarly as high (168.3 ± 18.6 µgN/L) as in the upper section

(Figure 2b). In the spring high-flow period, nitrate N concentrations decreased by 2.6 times to 65.4 ± 14.3 μgN/L and, in the summer–fall period, reached 54.9 ± 11.6 μgN/L. In contrast, this section was characterized by an increase in the concentration of ammonium compared with the upper section (Figure 2d). Nevertheless, the concentrations of total dissolved inorganic nitrogen in the river water of the middle section tended to decrease compared to the upper section, and the N:P molar ratio increased throughout the hydrological periods (Table 3).

### 3.4. Nutrients in the Lower Section

The phosphate concentrations remained similar when compared to the middle section, ranging from 12.4 ± 3.8 μgP/L in the summer–fall season to 18.0 ± 5.1 μgP/L in the winter low-flow period. There was an apparent gradual increase in the concentrations of phosphates and nitrate nitrogen observed from November to May. The winter concentrations of nitrate nitrogen demonstrated a decrease towards northern latitudes at the Igarka station (to 145.7 ± 57.4 μgN/L). Its concentrations sharply decreased in the spring period to 23.2 ± 31.1 μgN/L and remained low during the summer–fall period (21.2 ± 17.2 μgN/L) (Figure 2b). Downstream, the Arctic-GRO database reports even lower mean summer values for the Dudinka station (8.6 ± 7.7 μgN/L) (Figure 2b). Concentrations of ammonium nitrogen tended to decrease from the Zotino sampling station to Igarka and further to Dudinka in all hydrological periods (Figure 2d). Nitrite nitrogen was characterized by peak values (>10 μgN/L) at maximum discharges during the spring flood (Figure 2c), and its concentration at the Igarka station was significantly higher ($p < 0.01$) compared to the Zotino station.

The DIN concentrations in the lower section during the ice-free season dropped twofold (60.6–62.9 μgN/L) in comparison to the middle section (125.8–151.6 μgN/L), but showed no statistically significant changes in the winter low-flow period (Table 3). The N:P molar ratio ranged from 10 to 13 during the ice-free season and up to 25 in the winter low-flow season, which was significantly lower than in the middle section near the Zotino station. The Arctic-GRO database reports much less DIN and a much lower N:P ratio during the summer–fall period, while the spring and winter season N:P ratio values were somewhat higher relative to our data obtained for the Igarka station (Table 3). This fact was due to the lower ammonia levels reported in the database (Figure 2d).

### 3.5. Nutrient Loads

Despite the observed 6.7-fold increase in the mean annual discharge of the Yenisei River from the Bazaikha station to the Igarka station, the annual fluxes increased only threefold for P-PO$_4$ (from 2500 to 8100 tons/year) and twofold for N-NO$_3$ (from 13,650 to 28,240 tons/year) (Table 4). In contrast, the nitrite and ammonium loads increased more significantly by seven- and tenfold, respectively.

In terms of seasonal flux patterns, the lower-section winter flux was about 40% of the annual values for all forms of nitrogen and greater than 50% of the values for phosphates. The share attributable to the spring flood varied from 11 (nitrite nitrogen) to 30% (ammonium nitrogen) (Table 4, Figure 3). For the downstream stations, the contribution of the spring flood to the annual flux played a more significant role, which was especially evident for nitrite nitrogen (78% of the annual value at the Igarka site) and phosphates, the flux of which during this period reached 51% in the Bor area and 43% in Igarka. In the summer–fall period, the flux of inorganic forms of nitrogen and phosphorus was lower than in other periods of the hydrological year, despite its relatively long duration and large runoff.

**Table 4.** Seasonal and annual fluxes of nutrients at the gauging stations of the Roshydromet network: Bazaikha (Krasnoyarsk), Bor (Podkamennaya Tunguska) and Igarka.

| Period | Bazaikha | Bor | Igarka |
|---|---|---|---|
| $P-PO_4$, $\times 10^9$ g | | | |
| Spring flood | $580 \pm 170$ | $1630 \pm 430$ | $3240 \pm 600$ |
| Summer–fall | $590 \pm 130$ | $840 \pm 150$ | $2270 \pm 190$ |
| Winter low-flow | $1330 \pm 350$ | $740 \pm 220$ | $2590 \pm 120$ |
| Annual | 2500 | 3210 | 8100 |
| $N-NO_3$, $\times 10^9$ g | | | |
| Spring flood | $2350 \pm 390$ | $4390 \pm 1530$ | $5140 \pm 2720$ |
| Summer–fall | $5470 \pm 360$ | $1720 \pm 1080$ | $3300 \pm 670$ |
| Winter low-flow | $5830 \pm 810$ | $12,840 \pm 3150$ | $19,800 \pm 1260$ |
| Annual | 13,650 | 18,950 | 28,240 |
| $N-NO_2$, $\times 10^9$ g | | | |
| Spring flood | $50 \pm 40$ | $590 \pm 310$ | $2070 \pm 490$ |
| Summer–fall | $200 \pm 50$ | $260 \pm 120$ | $560 \pm 100$ |
| Winter low-flow | $190 \pm 70$ | $200 \pm 50$ | $380 \pm 50$ |
| Annual | 440 | 1050 | 3010 |
| $N-NH_4$, $\times 10^9$ g | | | |
| Spring flood | $760 \pm 200$ | $10,930 \pm 4450$ | $8340 \pm 2130$ |
| Summer–fall | $670 \pm 250$ | $5380 \pm 2030$ | $7820 \pm 1270$ |
| Winter low-flow | $1110 \pm 450$ | $5100 \pm 3260$ | $8140 \pm 1040$ |
| Annual | 2540 | 21,390 | 24,300 |

The nutrient yields (nutrient flux normalized to specific drainage area) obtained for the sampling stations demonstrated more complex spatial behavior for the analyzed compounds. The annual phosphate yield changed from 8.7 mgP/m$^2$/year in the upper section to 1.8 mgP/m$^2$/year in the middle section and 3.3 mgP/m$^2$/year in the lower-river section. The annual nitrate yield ranged from 47.4 mgN/m$^2$/year in the upper section to 11.6 mgN/m$^2$/year in the lower section. The maximum annual ammonia yield occurred in the middle section of the river, reaching 12.1 mgN/m$^2$/year, while in the upper section and lower section, the yields were 8.8 and 10.0 mgN/m$^2$/year, respectively. In contrast, higher nitrite yields were observed in the upper and lower sections at 1.5 and 1.2 mgN/m$^2$/year, respectively. The annual DIN yield showed a tendency to decrease with latitude, from 75.5 to 42.7 mgN/m$^2$/year.

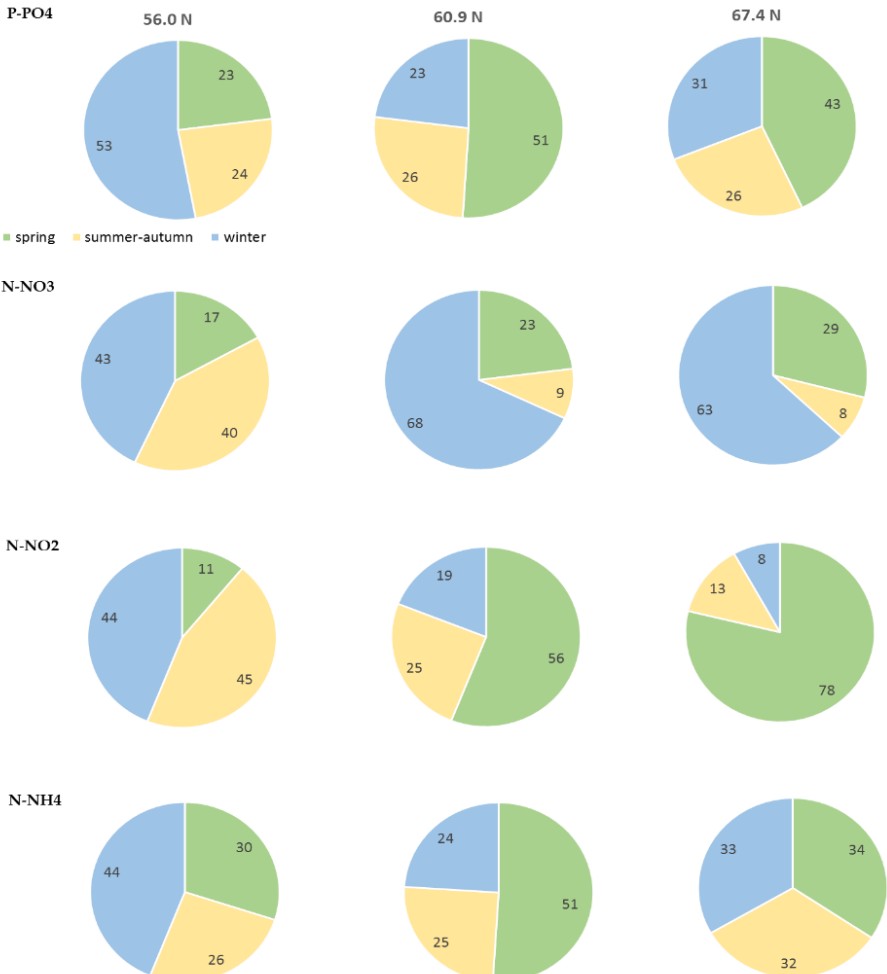

**Figure 3.** Relative seasonal distribution (%) of annual nutrient loads at different sections of the Yenisei river.

## 4. Discussion

Nutrient loads from terrestrial ecosystems to a drainage network are controlled by multiple biogeochemical and hydrological processes, as well as the inputs of biogenic elements from the atmosphere to the river basin area. In particular, atmospheric waters in the Yenisei basin contain sizeable amounts of phosphate P (from 7.3 μgP/L in snow to 18.7 μgP/L in rainwater) and nitrate N (from 130 μgN/L in snow to 185 μgN/L in rainwater). Further, river ecosystems process nutrient inputs from a drainage basin and produce new matter in dynamic ways that change seasonally and during downstream transport in river networks [50]. Studies of nutrient concentration in the hydrochemical composition of river runoff have shown that the waters of the Yenisei River near Krasnoyarsk are relatively rich in biogenic elements. The concentrations of inorganic phosphorus obtained in our study in 2015 to 2021 were in concordance with the ranges of values reported in earlier studies [45,51] for this upper section of the river (4–22 μgP/L). Similar findings have also been obtained for the unregulated upper section, where phosphate content does not exceed 5 μgP/L [52]. Nevertheless, some higher levels of biogenic compounds were demonstrated by Driukker et al. [42] and extremely high concentrations of phosphates, reaching 625 μgP/L, were found for the downstream section below Krasnoyarsk city [52], which the authors attributed to the input of effluents of waste waters. However, in our recent study, we did not observe such extremely high phosphorus concentrations in the Yenisei River within the upper section from the Krasnoyarsk dam to the Angara River [53].

Along with increased levels of phosphates, this section is characterized by elevated concentrations of inorganic forms of nitrogen, with the dominance of nitrate N. Similarly high levels of inorganic N were previously reported in the waters of the Krasnoyarsk reservoir [54,55]. A comparative study of the hydrochemical composition of the Krasnoyarsk reservoir waters and the Yenisei River in the lower reaches (i.e., downstream of the reservoir) suggested that the downstream river water originates from the deep layers of the reservoir [52]. Furthermore, our data demonstrating the minor seasonal variability in nutrients concentrations in the upper section corroborates these findings. Thus, it can be assumed that the high nutrient loads observed in the upper section of the Yenisei River (56–58° N) are controlled by biogeochemical processes regulating the hydrochemical composition of the waters in the reservoir. In particular, ongoing organic matter decomposition and nutrient release in the sediments of the reservoir may be the sources of elevated nutrient concentrations, as reported earlier [56]. On the other hand, the large area covered with agricultural lands within the reservoir catchment might also be a considerable source of the nutrients observed in surface waters.

In the lower reaches of the Yenisei River, the nutrient levels in the river channel significantly drop, despite the high atmospheric input of nutrients as snow and rain water. These findings highlight, specifically in the spring flood, the strong abiotic and biotic control of soil over the losses in nutrients from the terrestrial domain throughout the entire Yenisei basin. In riverine systems, the nutrient content can further differ dramatically across the longitudinal continuum as a result of differences in source contributions and mixing [57]. In the case of the Yenisei River, the sharp changes in nutrient concentrations (ca. twofold decrease) in the summer–fall period were observed after the confluence with the Angara River [53]. Low concentrations of nitrate in summer low-flow periods have also been reported for other Siberian rivers. According to Sanders et al. [20], the waters of the Lena River contain 2.8–19.6 µgN/L of nitrates. Holmes et al. [39] showed nitrate concentrations of 11.9 µgN/L in the Lena River, 35.1 µgN/L in the Ob' River and 14.7 µgN/L in the Kolyma River. For Western Siberian rivers, the values of the nitrate concentrations ranged from 2.5 µgN/L in Ob', Irtysh and Ket' to 9.9 µgN/L in the Tom' River [17]. On the other hand, there are significant variations in earlier estimates of nutrient concentrations in downstream sections on spatial and temporal scales. According to Bessudova et al. [44], the nitrate N content in the lower section of the Yenisei River was 40–70 µgN/L and 80–130 µgN/L prior to entering the Kara Sea. Exceptionally high concentrations of ammonium were found for the waters of the Yenisei River for 1970 to 1980, when they ranged from 270 to 1350 µgN/L [58]. There were lower, but elevated, values (270–870 µgN/L) shown for the summer season of 2012 [45]. In contrast, significantly lower levels of ammonium (9–13 µgN/L) were reported by the Arctic-GRO project in Dudinka (69.2° N) [39].

The decrease in the concentrations of dissolved inorganic nitrogen and phosphorus, which are the main substrate factors determining the growth of primary production, in addition to dilution (especially in the spring high-flow period) with the waters of the Angara River, is most likely due to an increase in phytoplankton activity. The low productivity of phytoplankton in the upper section of the Yenisei River is due to the low temperatures of the discharged deep waters of the Krasnoyarsk reservoir and the partial death of phytoplankton when passing through the turbines of high-pressure hydroelectric power stations [59]. Due to the descent of the deep waters of the Krasnoyarsk reservoir, the influence of which can be traced downstream at a distance of up to 500 km, the water temperature in the summer does not exceed 10–15 °C [45]. However, the inflow of warmer waters from the Angara River leads to an increase in temperature by 5–10 °C, which improves the conditions for the development of aquatic organisms [52]. There is no doubt that the main role in these processes is played by the biotic factors of the environment, among which the leading position is occupied by the flow velocity, which naturally changes along the riverbed and largely determines the turbulent mixing regime, the accumulation of eroded material and many other physicochemical characteristics [60]. In this section of the Enisey River, the channel width increases (at least 2000 m), the flow velocity decreases (to 0.8–1.1 m/s) [61]

and the elevated nutrient supply creates favorable conditions for the development of phytoplankton and the activation of photosynthesis [52,62].

Similarly low nutrient levels and N:P ratios in the downstream sections of the Yenisei River during the summer seasons were reported earlier in several studies [44,45,63]. These findings suggest that, already in its middle section, the river system turns into a process of chemostasis [50]. According to Creed et al. [50], river systems can shift from hydrological integration to biogeochemical processing, which appears in the third- or fourth-order streams. As flows accumulate downstream, the contributions of catchment processes become overwhelmed by the influence of in-stream processes. In particular, the increase in the flux of biogenic elements observed along the Yenisei River continuum, despite the decrease in concentrations, suggests a balance between the nutrient uptake and release processes. In winter, however, changes in the concentrations of nutrients in the channel runoff of the Yenisei River throughout the entire latitudinal transect are insignificant. However, they appear at maximum levels, which, in contrast, indicates the inhibition of biological processes involved in nutrient uptake and the prevalence of either their release during sediment decomposition or input from groundwaters or from soils [17,64]. Similarly, a high level of nitrate N in the winter period was reported for another major Siberian river in the Lena River delta [20].

High nitrite N concentrations were observed in the Yenisei River at the lower section in the spring flood compared to the summer and winter low-flow periods. Nitrites are intermediate products of organic matter decomposition and associated processes of nitrification and denitrification. However, this nitrogen form has not been previously considered in riverine N flux calculations due to insignificant concentrations [43,65]. Nevertheless, according to our data, there is a steady tendency for increasing nitrite N concentrations during the spring flood in the section receiving the large volumes of waters (ca. 40% of annual runoff) from the permafrost-affected drainage basins (i.e., Podkamennaya Tunguska and Nizhnyaya Tunguska rivers). These permafrost-affected basins host organically rich soils [4] and supply the Yenisei River with high loads of dissolved and particulate organic matter [46]. Our ongoing studies of rivers in the Nizhnyaya Tunguska River basin also demonstrate increased nitrite concentrations during the spring flood (Tokareva, unpublished data). The formation of a soil pool of nitrite N in winter is probably due to the presence of ammonium and dissolved oxygen ions, which lead to high nitrifying bacteria activity [66,67]. Simultaneously, the high content of ammonium and oxygen in soils explainen [66] the accumulation of nitrite ions in winter conditions, which are then released in the spring flood.

Along with ambient nutrient concentrations in aquatic ecosystems, the control exerted by the N:P ratio over primary production, nutrient cycling and resource competition is key. An N to P ratio equal to 16:1 has been shown to be an optimal stoichiometric proportion for these elements in aquatic systems [68]. The lowest N:P ratios were observed during the summer period, reaching 13 at the Igarka and 6 at the Dudinka stations at the lower sections of the Yenisei River. The N:P ratios are significantly higher than 16 in the upper and middle sections of the Yenisei River, as well as in the winter period; it can be assumed that these systems are limited in phosphorus. In contrast, in the high-latitude section (Igarka, Dudinka stations), the low N:P ratios reflect the increasing nitrogen limitations.

The Yenisei River accounts for up to 45% (636 km$^3$/year) of the total flow of the rivers (1350 km$^3$/year) flowing into the Kara Sea and, accordingly, it is a significant source of nutrients entering the Arctic Ocean [69]. The annual flux of phosphates increased from the Bazaikha area to Igarka. The nutrient flux for the Dudinka site was somewhat higher than that obtained for the Igarka site. In the work of Holmes et al. [39], the phosphate runoff in the Dudinka area (69.2° N) was estimated at $10 \times 10^9$ gP/year and nitrate nitrogen as $49 \times 10^9$ gN/year. During the period of natural runoff (the 1960s, before the construction of the Krasnoyarsk hydroelectric power station), the phosphorus flux was $0.5 \times 10^9$ gN/year in the upper reaches of the river and $3.0 \times 10^9$ gN/year in the lower reaches of the integral section [70]. For the 1980s, these authors showed that the phosphorus inputs in the Bazaikha

section were slightly lower at $0.4 \times 10^9$ gP/year, while, in the Igarka section, in contrast, it increased with $4.9 \times 10^9$ gP/year. At the same time, according to Sorokovikova et al. [52], in the area of wastewater discharge in Krasnoyarsk in the 1970s to 1980s, the phosphorus flux was $3.9 \times 10^9$ gP/year.

The total mineral nitrogen export (DIN = $NO_3$ + $NO_2$ + $NH_4$) in the region of Krasnoyarsk, according to our estimates, is $17 \times 10^9$ gN/year, reaching $56 \times 10^9$ gN/year at the Igarka section. Similar values (excluding nitrite nitrogen) are given in the work of Holmes et al. [39]: $51 \times 10^9$ gN/year. However, these values are mainly determined by the flux of nitrate nitrogen (96%), while in our studies, its share is significantly lower (51%). In the work of Sorokovikova et al. [52], for the period from 1970 to the 1980s, the authors give significantly higher indicators of the total mineral nitrogen flux: 47 and $99 \times 10^9$ gN/year for the upper and lower sections of the Yenisei River, respectively. The relatively low values obtained in our study were probably due to lower water discharge for the 2015–2020 time period ($542 \pm 61$ km$^3$/year) relative to the Arctic-GRO data (636 km$^3$/year). The total nitrogen export, including its organic form, is, according to the Arctic-GRO data [39], $163 \times 10^9$ gN/year.

## 5. Conclusions

The Yenisei River channel starts to act as a "chimney" system, processing the terrestrial matter, after the confluence with the Angara River and further downstream toward the Kara Sea. Analysis of data for the six-year observation period revealed the spatiotemporal heterogeneity in the nutrient loads, which was dependent on the geographical location and the hydrological period. The waters of the upper river section demonstrated significant and steady nutrient enrichment throughout the hydrological year. Our findings clearly reflect the influence of the Krasnoyarsk reservoir on the nutrient loads in the Yenisei River in its upper section up to the confluence with the Angara River. During the winter season, nutrient levels in the middle and lower river sections remain high and similar to the upper section. At snowmelt, and especially the summer–fall seasons, all biogenic element concentrations drop dramatically after the inflow of the Angara River, suggesting strong biological uptake and a shift of the river system to chemostasis. Specific N cycle processes in permafrost soils result in an increase in nitrite N in lower section of the Yenisei River during spring floods.

**Author Contributions:** I.V.T.: Nutrient laboratory analysis and interpretation, conceptualization, writing of manuscript draft; A.S.P.: Conceptualization; writing of manuscript draft, supervision, review and editing. All authors have read and agreed to the published version of the manuscript.

**Funding:** This work was supported by State Assignment no. 0287-2021-0008 and the Tomsk State University Development Programme ("Priority-2030"). The studies in the Zotino station were supported by RSF grant no. 18-17-00237-P (A. Prokushkin).

**Data Availability Statement:** Data used in this study are duly available from the first authors on reasonable request.

**Acknowledgments:** The authors gratefully acknowledge the significant efforts and invaluable assistance during the field campaigns provided by our colleagues: S. Titov, N. Sidenko, R. Kolosov and others. We express our deep gratitude to the reviewers for their time and work improving our manuscript.

**Conflicts of Interest:** The authors declare no conflict of interest.

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
