# Peer review of "Seasonal and Spatial Variability of Dissolved Nutrients in the Yenisei River"

_water, doi:10.3390/w14233935_

Round 1

Reviewer 1 Report

The Authors studied the content of various nutrients in a Siberian river at different latitudes for a long period (2015-2021). These kind are very useful for evaluation of the consequences of climate changes and anthropic interventions. 
The presented paper is interesting and it is evident the hard work of the authors. The study is well designed, the determinations are made accurately. Surely the obtained information is indicative for experts and scholars of the sector.

just a curiosity: why isn’t there at least one table showing the data obtained for each single year? In my opinion it will be interesting.

please tell me 

1) what is mio at line 93?

2) is it a citation at line 373 (production- hydro biological…. 1993)?

please attention to typos:  i.e. chemostatasis lines 378 and 459

The study is useful and interesting, I think that it could deserve the pubblication

Author Response

  1. Just a curiosity: why isn’t there at least one table showing the data obtained for each single year? In my opinion it will be interesting.

Answer: We are grateful for the suggestion. In this manuscript, we did not aim to demonstrate the interannual variability. However in our ongoing research we are indeed focused on such analysis. So this task is one of our priorities for future research.

  1. What is mio at line 93?

Answer: mio – million. For a better perception, we replaced mio with a million.

Corrected version: «Its average long-term runoff is 636 km3/year, its catchment area is 2.54 million km2….»

  1. Is it a citation at line 373 (production- hydro biological…. 1993)?

Answer: Yes, this is a link to a book from the References.

  1. Production and hydrobiological research of the Yenisei. / ed. G. I. Galaziy, Novosibirsk, Nauka, 1993, 195 p.

  1. Please attention to typos: i.e. chemostatasis lines 378 and 459

Answer: thanks a lot for pointing out the typos. We have corrected them in the text.

We express our deep gratitude to reviewer for his time and work on the improving our manuscript.

Reviewer 2 Report

I have two short questions to address in the revision and one comment:

Q1: On line 56, what is Pg

Q2: On line 93, what is mio?

C1: The extremely high P levels are concerning. This research is important in identifying this pollutant flowing into the Arctic.

Author Response

  1. line 56, what is Pg.

Answer: Pg is petagram, 1*1015 g. https://www.unitconverters.net/weight-and-mass/petagram-to-gram.htm

To improve understanding for readers, we have written the full unit of measure in the text.

Corrected version: «On the other hand, the actual N stock in permafrost-affected soils could reach 130 Pg (petagram) N (Voigt et al. 2020)»

  1. line 93, what is mio?

Answer: mio – million. For a better perception in the article, we replaced mio with a million. «Its average long-term runoff is 636 km3/year, its catchment area is 2.54 million km2….»

  1. The extremely high P levels are concerning. This research is important in identifying this pollutant flowing into the Arctic.

Answer: thank you for the comment. We intend to continue further studies of nutrient concentrations in the Yenisei River channel to identify its long-term dynamics as well to assess the fate of nutrients in Yenisei Gulf.

We express our deep gratitude to reviewer for his time and work on the improving our manuscript.

Reviewer 3 Report

#The article is well described, presenting consistent methodology and results.

#The article could be more interesting if the sampling sites were delimited in sub-basins or micro-basins (suggestion). Perhaps the discussion would be more comprehensive.

#Discussions about land use and occupation could have been placed (% agriculture, % forests, etc)

Author Response

  1. The article is well described, presenting consistent methodology and results.

Answer: Thank you for such appreciation of the our work!

  1. The article could be more interesting if the sampling sites were delimited in sub-basins or micro-basins (suggestion). Perhaps the discussion would be more comprehensive.

Answer: Thank you for the comment. In this study, we have divided the river basin into 4 parts/sub-basins (Materials and Methods section). For each section of the river we have presented results on the specific nutrient yield, but indeed not characterized qualitatively the gains or losess of nutrients in each section. In our further spatial analysis of nutrient fluxes in the Yenisei River basin we will focus on that task.

  1. Discussions about land use and occupation could have been placed (% agriculture, % forests, etc)

Answer: We are thankful for this suggestion. It is very important. Such data for the catchment area of the Yenisei River were collected during the manuscript preparation. However, on the later stage we have decided to skip this information and prepare a separate work with extended nutrient data set (including the tributaries of the Yenisei River having the stronger signal of landclasses) focused on the influence of peatlands, forests, agriculture etc.

We express our deep gratitude to reviewer for his time and work on the improving our manuscript.